# Effective Thermal Diffusivity Measurement Using Through-Transmission Pulsed Thermography: Extending the Current Practice by Incorporating Multi-Parameter Optimisation

**DOI:** 10.3390/s25041139

**Published:** 2025-02-13

**Authors:** Zain Ali, Sri Addepalli, Yifan Zhao

**Affiliations:** Faculty of Engineering and Applied Sciences (FEAS), Cranfield University, Cranfield MK43 0AL, UK; p.n.addepalli@cranfield.ac.uk (S.A.); yifan.zhao@cranfield.ac.uk (Y.Z.)

**Keywords:** infrared thermography, through transmission, thermal diffusivity, full-factorial design, finite element analysis, pulsed thermography

## Abstract

Through-transmission pulsed thermography (PT) is an effective non-destructive testing (NDT) technique for assessing material thermal diffusivity. However, the current literature indicates that the technique has lagged behind the reflection mode in terms of technique development despite it offering better defect resolution and the detection of deeper subsurface defects. Existing thermal diffusivity measurement systems require costly setups, including temperature-controlled chambers, multiple calibrations, and strict sample size requirements. This study presents a simple and repeatable methodology for determining thermal diffusivity in a laboratory setting using the through-transmission approach by incorporating both finite element analysis (FEA) and laboratory experiments. A full-factorial design of experiments (DOE) was implemented to determine the optimum flash energy and sample thickness for a reliable estimation of thermal diffusivity. The thermal diffusivity is estimated using the already established Parker’s half-rise equation and the recently developed new least squares fitting (NLSF) algorithm. The latter not only estimates thermal diffusivity but also provides estimates for the input flash energy, reflection coefficient, and the time delay in data capture following the flash event. The results show that the NLSF is less susceptible to noise and offers more repeatable values for thermal diffusivity measurements compared to Parker, thereby establishing it as a more efficient and reliable technique.

## 1. Introduction

Industry 4.0 is characterised by the advancement of various technologies such as the connectivity of computer systems, the Internet of Things (IoT), Big Data, Smart Factories, autonomous robots, predictive maintenance, and Additive Manufacturing. In the context of the non-destructive testing (NDT) sector, this translates into changes in inspection procedures driven by the progression of Industry 4.0 [1]. Traditionally, NDT is a manual procedure conducted to ascertain a material’s structural integrity in the presence of defects and damage. Advancements in technologies such as machine learning and deep learning have become a key aspect in interpreting complicated inspection data to achieve higher levels of detectability and overall efficiency of defect detection [2]. NDT offers a wide range of inspection technologies operating on a set of physical principles that determine structural anomalies within a component [2]. For example, ultrasonic testing, or UT, is a popular NDT technique that uses guided ultrasonic waves to locate and evaluate potential flaws. However, every NDT technique has its limitations. Some, such as X-ray inspection, rely on radiative methods, raising safety concerns, while others require the application of additional materials, such as water, gels, or magnetic particles, making them unsuitable for certain inspection scenarios. Infrared Thermography (IRT) has emerged as a non-contact, non-intrusive technique that overcomes many of these challenges, offering an alternative for defect detection [3].

The last two decades have seen rapid advancements in various IRT technologies to assess materials that are used for a wide range of applications. Pulsed thermography (PT), a subset of IRT, is a robust technique, making it an attractive solution to assess metals and composites [4,5]. A range of PT applications can be found in industries such as automotive, aerospace, defence, and oil and gas [6,7,8,9]. PT measurements can be taken in two modes: reflection and transmission. The key difference between these modes is the placement of the heat source and the IR radiometer with respect to the component under inspection. The reflection mode has both the radiometer and the heat source placed on the same side of the specimen, and the transmission mode has the heat source and radiometer placed on opposite sides.

A limitation of the reflection mode is that it loses its reliability in estimating thermal diffusivity when the thickness of a sample is over 3 mm [8]. This is problematic, as accurately determining the property is essential for evaluating the material’s structural integrity. Thermal diffusivity governs how quickly heat can diffuse through a material due to an induced temperature difference at the surface of a material. To combat this issue, multiple signal processing algorithms have been developed, such as Thermographic Signal Reconstruction [10], Pulsed Phased Thermography [11], Principal Component Thermography [12], Differential Absolute Contrast [13], Absolute Peak Slope Time [14], and Nonlinear System Identification [15]. Furthermore, keeping in line with the rapid progression of technologies in Industry 4.0, various AI technologies have been developed, such as forward-feeding neural networks and deep neural networks, to identify and classify defects caused by different environmental effects [9,16,17,18]. One other technique that has shown potential in accurately measuring thermal diffusivity is the through-transmission technique. Due to its higher levels of reliability, it has been commercialised into systems such as the Netzsch Laser Flash Apparatus (LFA) [18].

The literature shows that the primary motivation for developing PT is to enhance its defect characterisation capabilities. The introduction of a defect within a material, such as an air gap, results in a time delay for heat propagation through its thickness due to the significantly lower thermal conductivity of air. This thermal delay translates into a localised hotspot which can be utilised to identify and characterise a defect. For PT, the majority of the research has been focused solely on the reflection mode [19]. A major reason for this is the transmission mode’s inability to quantify defect depth despite offering greater lateral resolution for defects and the ability to detect deeper subsurface defects when compared to the reflection mode, as demonstrated by Maierhofer et al. [8]. This has prompted researchers to refocus their efforts solely on the defect detection capabilities of through-transmission thermography by exploring the 2-D spatial distribution of thermal diffusivity [20]. However, a recent review indicated that while the technique has a lot of potential, it has not been explored as much as the reflection mode [19]. A plausible explanation for this is attributed to the fact that for the transmission mode, the distance travelled by the thermal waves in both the defect and defect-free zones is the same [21]. However, this explanation provides limited evidence mathematically or theoretically to support the claim. The authors of this study have previously attempted to provide a more in-depth analysis to address this knowledge gap by developing a finite element model using COMSOL (v.6.0) [22]. The paper implemented a Taguchi Design to analyse the thermal contrast, i.e., the temperature difference between the sound and defect areas. It was concluded that the peak thermal contrast is more sensitive to defect size than depth. Furthermore, it was determined that the signal-to-noise ratio (SNR) is less sensitive to defect depth when compared to the reflection mode [22]. However, after a certain depth, as the defect moves closer to the back surface, the transmission mode offers a better SNR compared to the reflection mode [19].

Whilst there have been numerous studies that explore the effect of different input parameters on the detectability of defects [23,24,25,26,27], they lack a comprehensive illustration of their influence on the defect detection limits. Only recently, a comprehensive study was published that explored the defect detection limits using a wide range of inspection conditions [28]. However, this study illustrated the limitations of the reflection mode exclusively using a simulation environment. This was due to the extensive nature of the study, which would take years to replicate physically. The present literature continues to show a lack of studies exploring the capabilities of the transmission mode in defect detection, even though it is a well-established technique for determining the thermal diffusivity of various materials [29,30,31,32]. Furthermore, research also shows a lack of studies employing techniques such as factorial designs to understand the interaction of multiple variables in the calculation of material thermal properties such as thermal diffusivity. It must be understood that material thermal diffusivity is an intrinsic property of any given material and does not change with its geometry. However, it has been demonstrated previously that practical measurements will adversely affect the measurement uncertainty in the thermal diffusivity measurement for the same material when input variables such as sample thickness and flash energy are varied [33,34].

Previous work by the authors highlighted the existing limitations of active thermography [19], particularly the absence of theoretical models that simulate PT in transmission mode. Moreover, it showed that the existing literature does not contain a simple step-by-step procedure for conducting through-transmission PT inspections. This study aims to address these two issues to help mitigate some of the existing challenges hindering the advancement of through-transmission PT [19]. Here, the effect of varying thicknesses and flash energy levels in thermal diffusivity evaluation is analysed. These variables were chosen due to their well-documented influence on thermal diffusivity estimation, as mentioned above. Furthermore, they are fundamental to the technique and relatively straightforward to manipulate in controlled experiments: flash energy determines the heat input to the sample, while sample thickness governs the thermal response dynamics and heat propagation, both of which directly impact the thermal response signal critical for diffusivity estimation. In contrast, factors such as environmental conditions (e.g., ambient temperature and air currents), sensor characteristics (e.g., quantisation and noise), and system calibration are more challenging to control or to vary systematically, often introducing confounding effects that complicate analysis. Moreover, in laboratory settings, the environment is relatively consistent; therefore, its effect on thermal diffusivity estimation is negligible. Consequently, flash energy and sample thickness offer a practical and precise framework for investigating the effects on thermal diffusivity estimation. Also, previous research has identified these two factors as the most influential in accurately assessing thermal diffusivity [34]. Moreover, this study also seeks to update the current established practice of thermal diffusivity estimation by incorporating multi-parameter optimisation. Additionally, the sensitivity of the camera standoff distance to the sample in evaluating thermal diffusivity is also investigated. Since there are only two variables that are being manipulated (sample thickness and flash energy level), a full-factorial design of experiments (DOE) is considered. The justification for using this type of DOE compared to other more practical design of experiment methods, such as Taguchi design, is because a full-factorial DOE offers a more comprehensive understanding of the interaction of multiple parameters on the output. Moreover, Taguchi is preferred when there are multiple input parameters leading to a large set of experimental runs. However, the number of runs in this study is not high enough to render the full-factorial design impractical.

The intended outcome of this research is to provide a standardised approach to measuring thermal diffusivity using flash thermography in the through-transmission configuration. While commercial apparatus such as the Netzsch LFA exist, they are expensive due to the inclusion of temperature-controlled chambers and the need for multiple calibrations. Furthermore, they often require samples to be of specific dimensions for measurement. The setup in this paper will provide a simple and more cost-friendly alternative that can be replicated in laboratory settings. The novelty of this paper lies in its assessment of the viability of thermal diffusivity measurement using PT in the through-transmission configuration using multi-parameter optimisation together with Parker’s half-rise equation, providing a comprehensive understanding of all the parameters that have an impact on the measurement (see Section 3.4 for more details). Furthermore, the scope of this research aligns with identifying the applicability of the through-transmission technique for assessing advanced materials used in fields such as additive manufacturing by providing a turnkey solution.

## 2. Theory

This section discusses the fundamental theory that governs heat transfer in a material. It is critical that the fundamental theory be fully understood to appreciate the strength of the through-transmission technique, something that has not been extensively documented by researchers. Based on this, the equations governing the front and back wall temporal behaviour of a sample exposed to pulsed heating are derived. From those equations, the thermal diffusivity of any material can be measured.

### Pulsed Thermography (PT)

For PT, the heat source comprises either a laser source or flash lamps that deliver an instantaneous pulse of heat lasting in the order of milliseconds. The applied heat is absorbed by the specimen’s surface, thereby increasing its temperature. The theory of heat diffusion follows the classical heat diffusion equation, also known as the Fourier equation, which is as follows:(1)$$\frac{\partial T}{\partial t}=\alpha \nabla^{2}T$$
where $\alpha =\frac{k}{\rho C}$ is the thermal diffusivity of the material (m^2^/s), k is the thermal conductivity (W/mK), ρ is the density (kg/m^3^), C is the specific heat (J/kg K), T is the temperature (K), and t is the time (s). Fourier’s law of heat conduction states that the rate of heat transfer through a material is proportional to the negative gradient in the temperature and to the area, at right angles to that gradient, through which the heat flows. Since the heat is applied at the front surface, the negative temperature gradient is highest in the thickness direction, making it the dominant direction in which heat flows. Moreover, the lateral dimensions of the sample are significantly larger than the thickness, resulting in negligible negative temperature gradients along those directions despite the isotropic nature of thermal properties in homogeneous materials. This reduces the three-dimensional heat transfer problem to a one-dimensional heat flow, allowing Equation (1) to be simplified as follows:(2)$$\;\frac{\partial T}{\partial t}=\alpha \frac{\partial^{2}T}{\partial x^{2}}\;$$
where x is the direction parallel to the through thickness of the material. While there are numerous solutions for the equation above with different boundary conditions, a popular solution was determined by Carslaw and Jaegar [35]. The equation assumes a semi-infinite plate that is thermally insulated. The initial temperature is defined as $T\left(x,0\right)$, and the temperature later on at any given time *t* can be defined as follows:(3)$$T\left(x,t\right)=\frac{1}{L}\int_{0}^{L}T\left(x,0\right)dx+\frac{2}{L}\sum_{n=0}^{\infty }e^{\left(\frac{-n^{2}\pi^{2}\alpha t}{L^{2}}\right)}\times \cos\left(\frac{n\pi x}{L}\right)\int_{0}^{L}\frac{T\left(x,0\right)\cos n\pi x}{L}dx\;\;$$

Parker et al. [31] took the solution derived by Carslaw and Jaegar and added specific conditions to overcome two key challenges to solving Equation (3). First, the thermal contact resistance between the sample and the heat source is eliminated by introducing flash excitation. Second, the data capture time is reduced to the point where surface heat losses can be avoided. Applying these conditions, for a semi-infinite plate with length L subjected to a Dirac pulse of energy value Q, the temporal evolution of the front and back walls can be calculated using the following equations. For the front wall:(4)$$T\left(0,t\right)=\frac{Q}{\rho CL}\left[1+2\sum_{n=0}^{\infty }\exp\left({-n}^{2}\pi^{2}\frac{\alpha t}{L^{2}}\right)\right]$$
and for the back wall:(5)$$T\left(L,t\right)=\frac{Q}{\rho CL}\left[1+2\sum_{n=0}^{\infty }{\left(-1\right)}^{n}\exp\left({-n}^{2}\pi^{2}\frac{\alpha t}{L^{2}}\right)\right]$$
where ρ is the density (kg/m^3^), C is the material’s specific heat capacity (J/kg K), and α is the material’s thermal diffusivity (m^2^/s). The equations above can be made dimensionless to make the relation between temperature and time independent of material properties. This is done by introducing two dimensionless parameters. The first parameter is $V=\frac{T}{T_{M}}$, where V is the ratio of the temperature T to the maximum temperature $T_{M}$, and the second is $\omega =\frac{\pi^{2}\alpha t}{L^{2}}$, the dimensionless time quantity. Using this relation, Equation (5) is then reduced to the following:(6)$$V\left(L,t\right)=\left[1+2\sum_{n=0}^{\infty }{\left(-1\right)}^{n}\exp\left({-n}^{2}\omega \right)\right]$$

For the front wall, Equation (6) remains the same as the front wall Equation (4) but with the inclusion of the term ${\left(-1\right)}^{n}$ to indicate the direction of heat flow in the 1-D model. That term refers to the phase inversion of the thermal wave that has reflected off an adiabatic wall at the back causing a temperature rise. Figure 1 presents a schematic representation of the through transmission configuration with its respective back wall temperature curve based on Equation (6). Note that since V and ω are dimensionless quantities, the axis labels do not contain any units.

As seen in Figure 1a, the flash lamps are used to thermally excite the specimen from the front. On the other side, an infrared radiometer is placed at a certain standoff distance (See Section 4.2 for details) to record the back wall temperature.

Using the temperature curve at the back wall, Parker et al. determined that when the temperature at the back wall reaches half its maximum temperature, the value of ω=1.38, as shown by the dotted red lines in Figure 1b. If either the sample length or thermal diffusivity is known, the other value can be calculated using the following relationship:(7)$$\alpha =\frac{1.38{\;L}^{2}}{\pi^{2}t_{\frac{1}{2}}}$$

The above method, known as Parker’s half-rise equation, has become the gold standard method for measuring thermal material properties using PT and has also been employed by commercially available instruments. The thermal material properties evaluated using this method are for materials that are defect-free. For materials that contain defects such as air gaps, the temperature response will be different in the defect and defect-free zones. These temperature anomalies can be used to characterise the defects. Since these anomalies are caused by local thermal diffusivity variations within the material, it is crucial to establish a simple and repeatable approach that can effectively measure this value.

While other thermographic techniques such as lock-in, frequency-modulated thermography, and step heating exist, PT offers several advantages, such as improved SNR, faster inspection times, and the ability to operate without prior knowledge of specific modulation frequencies [36]. It has also been observed that longer heating times, such as the ones provided by lock-in and step heating, allow for the detection of deeper defects [37,38,39]. Moreover, recent studies have employed novel algorithms such as diffusion-compensated correlation analysis for frequency-modulated thermal signals to further reduce temporal noise, allowing for better defect resolution [40]. Finally, based on the available literature, a comparative study between pulsed thermography (PT) and step-heating thermography has demonstrated that step heating achieves a higher damage detection efficiency (94% compared to PT’s 88%). However, PT provides significantly more accurate defect size estimation, with an average error of 16.64%, whereas step heating results in a much higher average error of 37.73% [37]. This, combined with the previously mentioned advantages of PT, makes it a strong contender for material characterisation and, therefore, is the sole focus of this study.

## 3. Methods and Materials

The purpose of this section is to outline methods and materials used in the study to estimate thermal diffusivity. The methodology is applied to both the finite element analysis (FEA) and the laboratory experiments.

### 3.1. Materials

The samples used for the laboratory experiments are seven 100 mm × 150 mm S275 grade steel plates (thermal diffusivity = 14 mm^2^/s, based on the material information provided by the supplier) with thickness values ranging from 2.118 mm to 9.87 mm. The entire surface of the steel plates (front and back) is painted black with a matte finish in line with the previous works primarily to reduce sample reflectivity and to ensure that its emissivity values are close to 1 [34,41]. If the sample has low emissivity, the radiation detected by the radiometer will not accurately represent the sample’s temperature. While potential solutions to address this issue have been explored in the literature, including those presented by Ruffio et al. [42], they require multiple flashes, which would significantly increase the experimental time. Figure 2 shows the back wall temperature plot from Figure 1b at different emissivity values by multiplying different emissivity values with the output of Equation (6). It can be inferred from the figure that lower emissivity causes the back wall temperature to increase at a slower rate. This occurs because samples with lower emissivity absorb less incidental radiation on their surface and reflect a greater proportion of it.

For the thermal excitation, two optical flash lamps, each lamp providing a maximum excitation energy of 4.8 kJ, are used. A detailed overview of the thermal excitation values used for the laboratory experiments can be seen in the next subsection of this paper. The lamps are synced with the data acquisition software Research IR and the IR camera so that the frame at which the flash occurs can be recorded. The pulse duration of the lamps can be varied to achieve the required level of energy. According to the manufacturer, it offers 1/175 s at 4.8 kJ, 1/300 s at 2.4 kJ, 1/500 s at 1.2 kJ, 1/900 s at 0.6 kJ, and 1/1425 s at 0.3 kJ. The IR camera used is a FLIR SC7600 mid-wave camera (3–5 μm wavelength range) offering a resolution of 640 × 512 pixels operating at a frame rate of 100 Hz.

### 3.2. Finite Element Modelling

To simulate flash thermography, the “Heat transfer in Solids” physics module available in COMSOL (v.6.0) was used to model a steel plate with dimensions 150 × 100 mm and varying depth. Figure 3 shows the FEA model. To reduce the simulation time, only a small region (highlighted in yellow) is finely meshed. Here, the option of an extremely fine mesh is selected, which is a size preset in COMSOL. The size of the elements used in both the region of interest (extremely fine) and the rest of the geometry (normal) are shown in Table 1 (under the heading “Meshing Parameter (area of interest)” and “Meshing Parameter (rest of the geometry)”). The back wall temperature information is extracted by taking a surface average of the yellow region once the simulation is complete. The analysis comprises a parametric sweep that encompassed all the sample thicknesses at each energy level for a duration of 10 s with a time step of 0.01 s to replicate the rate of acquisition offered by the FLIR SC7600 IR Camera.

To ensure that the simulations are accurate, all the input parameters, material properties, and dimensions are chosen to be the same as the ones used in the experiments. For the flash heat, a heat flux was applied to the front surface at different energy levels. To control the pulse duration, a pulse function was created that simulates a square pulse *B*(*t*). The width of the square pulse is adjusted to set the pulse duration. The flash energy and duration are set to match the experimental conditions. The region of interest is set in the centre of the plate to be consistent with the region of interest selected for the experiments (see Section 4.2). The justification for using the centre of the plate is based on previous work found in the literature that indicates obtaining temperature information from this region provides the most accurate and reliable result [43,44]. Table 1 shows the input parameters used in COMSOL. The thermal properties and material geometry are selected in accordance with the steel plates that are used in the experiments. The inspiration for Table 1 is drawn from Vora et al. [45]. It is one of the few papers that outlines the exact input conditions used in COMSOL so that the study can easily be replicated. Since one of the objectives of this paper is to standardise the methodology through which data are collected in the transmission mode, there should be a repeatable approach outlined to do so, as mentioned previously by Ali et al. [19]. This approach should be clearly constructed so that the readers of this research can easily reproduce its contents.

### 3.3. Full-Factorial Design

Full-factorial design of experiments has been a popular choice that allows an investigator to study the effects of multiple input parameters on an output response. A detailed explanation of how to conduct a full-factorial DOE can be found in references such as Elser [46]. It allows for measuring the response of all combinations of the factors. In this study, the commercially available software Minitab (v.21.2) is used to set up the DOE and to evaluate the results. For flash thermography, the independent variables are the flash energy (kJ) and the sample thickness (mm). The dependent or the calculated variable is the thermal diffusivity (mm^2^/s). The value of the thermal diffusivity is calculated using two methods: Parker’s half-rise method [31] and the New Least-Squares Fitting method first introduced by Sirikham et al. [47]. Table 2 lists the number of runs along with the independent variables for flash thermography. The sample thickness is measured using a digital vernier calliper ten times and then averaged. These averaged values are the ones displayed in the table below.

### 3.4. Multi-Parameter Optimisation

As mentioned in Section 2, Parker’s half-rise equation still remains the most popular method to evaluate material thermal diffusivity. However, since it is only taking the time at which the temperature reaches half the maximum, it is prone to errors, as it does not take into consideration the heat loss at the back wall, which has been highlighted by Chihab [48]. Sirikham et al. introduced the NLSF method to better estimate the thermal wave reflection coefficient [47]. For a given temperature vs. time graph for a sample exposed to flash heating, the method uses the theoretical equation first developed by Lau et al. [49] that describes the temperature behaviour and estimates the unknown parameters using the built-in nonlinear least-square fitting *(lsqnonlin)* solver in MATLAB. While the NLSF algorithm has been used effectively in the reflection mode, its efficiency in determining the thermal diffusivity in the transmission mode has not been investigated. Instead of using just the half-rise time, the entire equation, i.e., Equation (5), is used. In the equation, four unknown parameters need to be estimated, which are the thermal diffusivity α, the input energy divided by the product of the material density and specific heat capacity $W=\frac{Q}{\rho C}$, the reflection coefficient *R*, and the start of sampling time *t_s_*. The addition of the data acquisition time is introduced to allow for using the segment of data that might start after the time *t* = 0, which is the time at which the flash event occurs. This addition is unique and has not been used in previous works such as Krapez et al. [50,51]. It is worth noting that these models were not applied to PT, specifically in the transmission mode. Additionally, it is important to understand and differentiate the flash energy and sample thickness parameters. To demonstrate the strength of the through-transmission PT technique, only flash energy and sample thickness have been considered, as they are independent variables that have maximum impact on the measurement. The parameters in the multi-parameter optimisation are dependent variables directly from Equation (5) that are being estimated resulting from variations in the independent variables. The proposed method finds the optimal parameters based on the following relation:(8)$$\underset{W,R,\alpha ,t_{s}}{\mathrm{argmin}}\left\|\tilde{T}\left(t\right)-T(t)\right\|$$

The selected initial value for each parameter is as follows: *α* is the theoretical material thermal diffusivity obtained from the manufacturer, i.e., 14 mm/s^2^; *R* is the reflection coefficient, which for the transmission mode would ideally be -1; *W* is determined by the input flash energy (9.6 kJ, 4.8 kJ, and 2.4 kJ); and *t_s_* is 0, assuming the data acquisition started right after the flash event. To ensure that optimisation does not diverge, lower and upper bounds were selected for each parameter. For W, the upper and lower bounds were set 5 times higher and 5 times lower than the initial value. For *R*, the boundaries were −1.5 and 1.5. For the thermal diffusivity, the value was set as 10 mm^2^/s and 20 mm^2^/s, and the boundaries for *t_s_* were −1 and 1. Using this function, Equation (5) is re-written as follows:(9)$$T\left(L,t\right)=\frac{W}{L}\left[1+2\sum_{n=0}^{\infty }{\left(R\right)}^{n}\exp\left({-n}^{2}\pi^{2}\frac{\alpha (t+ts}{L^{2}}\right)\right]$$

## 4. Results and Discussion

The following subsections present the results obtained in this study. First, the findings from the finite element model are introduced, followed by those from the laboratory experiments. Finally, a comparative analysis of both approaches is provided.

### 4.1. Finite Element Model

Figure 4 shows the evaluation of the thermal diffusivity. Based on the material thermal properties listed in Table 1 obtained from the material datasheet from the supplier, the theoretical material thermal diffusivity value is 13.47 mm^2^/s. The material thermal diffusivity is estimated using both the NLSF and Parker methods. The NLSF algorithm not only provides an estimate for the thermal diffusivity but also for the input energy *Q*, the reflection coefficient *R*, and the starting time of the sampling *t_s_* (the value is 0 for the simulation model as the flash is set to occur at that time). It must be noted that Parker’s half-rise method is only able to estimate the thermal diffusivity. From the Figure, it is observed that at all energy levels, the NLSF slightly overshoots the diffusivity value, while Paker undershoots it.

A detailed overview of the NLSF’s performance in its four-parameter optimisation is listed in Table 3. The table shows that the algorithm is most accurate in estimating *Q* and least accurate in estimating *R*. Nevertheless, the measurement error is below 5% for all parameters at all energy levels. Furthermore, it is also observed that the percentage error in estimating the thermal diffusivity decreases at 2.4 and 4.8 kJ of input energy. For 9.6 kJ, however, the more accurate estimations are for the thinner plates. Additionally, closely examining the standard deviation for all the estimated parameters reveals that for simulations, the least variation is observed for the input energy of 4.8 kJ.

Figure 5 presents the average thermal diffusivity measurements for both methods across all energy levels and sample thicknesses. Observing the measurement error in Figure 5, it can be seen that the NLSF offers more accurate estimations for thermal diffusivity at lower thicknesses (thickness ≤ 3 mm), with a percentage error of below 2%. Once the thickness of the sample increases above 3 mm, the estimation error of Parker falls below 1.5%, while that of NLSF remains between 1.5% and 2%. Another observation is that the error in NLSF is found to be more consistent and less susceptible to thickness variations. Moreover, it should also be highlighted that the NLSF algorithm is estimating four parameters compared to Parker’s, which assesses only the thermal diffusivity parameter. NLSF’s thickness-independent nature and its ability to estimate multiple parameters demonstrates its superiority to Parker. The advantage of using multi-parameter optimisation is that it accounts for parameters such as input energy (*Q*), reflection coefficient (*R*), and sampling time (*t_s_*), which are difficult to measure in practice. These parameters are crucial, as they influence the temperature–time plot of a sample subjected to pulsed heating. Notably, in PT, *t_s_* is assumed to be zero if data capture begins immediately after the flash. However, achieving this would require specialised equipment capable of synchronising the flash with the start of data capture. The use of NLSF algorithms eliminates the need for such equipment as it estimates the time it took after the flash for the data capture to begin.

To further illustrate the robustness of the NLSF algorithm, several degrees of noise were added to the temperature data obtained from the simulations. The reasons for adding noise to the raw temperature data are several, some of which are mentioned here. Firstly, unlike simulations, temperature data obtained from laboratory experiments have a certain degree of noise embedded within them. The presence of noise can stem from various factors, including
1.Sensor noise, where there is intrinsic noise generated by the IR sensor itself [52].2.Readout noise generated by reading the signal from the IR sensor [52].3.Amplifier noise that is added by the amplifiers in the signal processing chain [52].4.Noise caused by changes in the environmental temperature [53].5.High humidity levels affecting the readings, as water vapour can absorb and emit IR radiation that can adversely affect the readings [53].6.Quantisation noise that occurs during the quantisation of the analogue IR signal to digital signal [54].

From the list above, points 4 and 5 have been neglected in this study as the technique development is done in a laboratory environment where the humidity levels and temperature are relatively stable and will not produce any significant variation in thermal diffusivity estimation. Moreover, since the sample is matte painted, the environmental effects are also significantly reduced [53].

Secondly, it is important to understand the limitations of the algorithm. The presence of noise can significantly affect the performance of the curve-fitting, and therefore it is necessary to establish the threshold for the algorithm. This will help in identifying the limitations of the algorithm and the conditions under which it is effective. Finally, it serves as a validation technique that will determine whether the algorithm will perform its desired function in practical settings.

Figure 6 demonstrates the NLSF algorithm’s performance at different SNR levels, ranging from high SNR (40 dB) to low SNR (20 dB). The figure shows that even at extremely low SNR levels (e.g., −5 dB), the algorithm is able to predict the thermal diffusivity α (Figure 6a) and the input energy level Q (Figure 6b) with the error varying between 0% and 13% for the thermal diffusivity measurement and between 0% and 3% for the energy level measurement. The SNR had to be reduced to −20 dB to get significant variation in the accuracy of the algorithm. While the algorithm estimates four parameters simultaneously, the error analysis in Figure 6 has been limited to the thermal diffusivity and energy level measurements. This is because the reflection coefficient R follows a similar pattern to Figure 6a,b, and the sampling time *t_s_* has a negligible error (less than 1%) regardless of the noise level. This could be attributed to the fact that the simulation started at t=0 and, regardless of the noise level, the start time is always the same. This shows the robustness of the NLSF algorithm, making it a practical alternative for measuring the thermal diffusivity of materials as well as estimating multiple parameters for a pulsed thermographic inspection. The justification for choosing NLSF over Parker is further strengthened by Figure 7, where a comparative analysis of both methods’ accuracy in estimating thermal diffusivity at varying SNRs is presented. Our research demonstrates that Parker delivers more accurate measurements at a higher SNR. A critical analysis reveals that the technique offers larger variation from the mean value when compared to the NLSF algorithm. Further, the analysis also confirms that at a lower SNR (−20 dB), the NLSF exhibits a much lower error (≈60%) compared to Parker (≈1600%), as shown in Figure 7. It can be inferred from Figure 7 that the percentage errors decrease significantly when the SNR is 10 dB or higher. To provide a more detailed view of these errors, the insets within the red boxes offer a magnified view of each plot. These zoomed sections reveal that, as SNR increases, the Parker method tends to yield more accurate measurements of thermal diffusivity. Nonetheless, the NLSF algorithm also delivers comparable results across the range of SNRs. Additionally, the NLSF algorithm offers estimates for multiple parameters, including the input energy *Q*, reflection coefficient *R*, and sampling start time *t_s_*, in addition to thermal diffusivity. This comprehensive capability makes it a more versatile and appealing option for thermal diffusivity estimation.

Finally, one of the main reasons for conducting a full-factorial DOE is to identify the parameter that has the most sensitivity toward the thermal diffusivity measurement. The answer to this question was achieved by using the “Analyse Factorial Design” option in Minitab. The Pareto charts in Figure 8 show the effect that the input variables, i.e., the sample thickness and flash energy level, have on estimating the material thermal diffusivity. The variables are on the y-axis, and their corresponding standardised effects are on the x-axis. Standardised effects simply mean that the effects of the parameters are scaled based on their standard error, which allows us to compare the significance of the two factors. The vertical red dashed line at 2.179 represents the point where the significance level is 0.05. If the standardised effect of a factor exceeds that line, then it means that the factor is considered to be statistically significant. Figure 8a shows the effect of the thickness and energy level in estimating thermal diffusivity using the NLSF algorithm where it can be seen that the flash energy plays a significant role in influencing the outcome of the thermal diffusivity measurement. In contrast, the role of sample thickness is shown to be not statistically significant in this particular case, and, therefore, it did not play a major role in influencing thermal diffusivity measurement. Given that thermal diffusivity is supposed to be an intrinsic material property, the result is in line with what was expected. However, the outcomes are flipped in Figure 8b, where the sample thickness is shown to play a much more significant role in estimating material thermal diffusivity compared to the flash energy level when the measurement method is Parker’s half-rise equation (Equation (7)). It is also important to note that, in this case, both factors exceed the threshold for a standardised effect (indicated by the red vertical line at 2.179), demonstrating their significant influence on thermal diffusivity estimation. However, the effect of flash energy is smaller relative to that of thickness. This means that variations in thickness introduce greater variability into the thermal diffusivity estimates using Parker.

Combining the observations in Table 3 and Figure 8, it can be said that the NLSF algorithm would be the preferred method of choice as it is not sensitive to sample thickness, and a high energy level such as 9.6 kJ should be used to ensure that the error in thermal diffusivity measurement is minimised. Higher energy leads to a greater temperature change at the back wall, improving the SNR. This will lead to more accurate measurements of thermal diffusivity. The noise equivalent temperature difference (NETD) of the IR sensor dictates the extent of signal variation observed when the camera captures electromagnetic radiation from a temperature-stabilized flat-field target [55]. From a practical perspective, this aligns with the established understanding of an IR sensor’s sensitivity to temperature variations. Any temperature variation below this NETD is undetectable by the camera. Consequently, the detection threshold for an IR sensor, ∆*T*, as described by Shepard et al. [55], is defined as ∆*T* = *nσ*, where *σ* represents the NETD and *n* is a multiplication factor to ensure that ∆*T* is at least n times greater than the NETD. While a case could be made that Parker’s half-rise equation delivers more accurate results at higher thicknesses, as illustrated in Figure 5, the difference in the error values is not significant enough to write off the ability of the NLSF algorithm to estimate thermal diffusivity at higher thickness, as both methods deliver a negligible uncertainty level for practical applications.

The purpose of the residual plots in Figure 9 is to validate the assumptions for the regression model for thermal diffusivity measurements for both the NLSF (Figure 9a) and Parker (Figure 9b) methods. The term “residual” itself simply refers to the difference between the observed value and the value predicted by the model. For both approaches, the figure in the top left is the “normal probability plot”, which shows that the residuals are approximately normally distributed. The plot on the top right is the “residuals vs. fits” plot that indicates no obvious patterns, which suggests that the linearity and homoscedasticity assumptions are met. This means that the variance of the residuals is constant across all levels of fitted values. The histogram in the bottom left plot reiterates the normality assumption, as the results roughly display a symmetric distribution. The “residual vs. observation order” plot, displayed in the bottom right of Figure 9 shows that residual values are scattered randomly without any discernible patterns, as they are plotted against the order of data collection. This random distribution suggests that there is no autocorrelation present, affirming that the values are independent of one another. Such independence is crucial, because it validates the assumption in the regression analysis that the errors in the data collection and prediction processes do not influence each other, thereby supporting the reliability and accuracy of our model’s findings. Finally, it is observed that residual values are similar for both the NLSF algorithm and Parker’s half-rise method, indicating that the choice of method has negligible effect on the thermal diffusivity measurement.

### 4.2. Results from Laboratory Experiments

As mentioned in the previous section, the temperature rise at the back wall can be prone to heat losses that might negatively affect thermal diffusivity calculations. Furthermore, the sensitivity of thermal diffusivity to IR camera standoff distance from the sample is also subject to investigation. To address these challenges, a random sample (3.859 mm thickness) was chosen from the selection of samples and was subjected to flash heat of 9.6 kJ. The back wall temperature information was collected at three different standoff distances: 200 mm, 250 mm, and 300 mm. The justification for using these distances was to cover an entire range of distances that were deemed practical for data capture. At distances below 200 mm, the entire sample was not fully captured within the camera’s field of view, while at distances beyond 300 mm, the sample occupied a minimal portion of the frame. To reduce random noise generated by the IR signal, a total of 10 repeats were conducted and averaged. For each run, the temperature measurements were recorded at the back wall for a total of 10 s with a region of interest (ROI) set to a 100 × 100-pixel square at the centre of the sample to negate thermal losses that might result from edge effects, which negatively affect the results [43,44] (at a standoff distance of 300 mm, each pixel corresponds to approximately 0.4 mm). This was done to ensure that temperature saturation, i.e., the maximum temperature that the back wall can reach, was within the captured range. To ensure that the captured dataset was indeed repeatable, the *p*-value estimate obtained via an analysis of variance (ANOVA) was determined using the built-in “data analysis” tool available in Microsoft Excel. If the *p*-value is less than α = 0.05, then there is a significant deviation between the results. For comparison, if each of the repeats had the exact same value, the *p*-value would be 1. When all the 10 s of data were considered for the repeats, the *p*-value was significantly less than 0.05. To determine the appropriate data segment, two conditions were set that needed to be satisfied in order to achieve a more accurate measurement. First, temperature saturation must have occurred at the back wall, and second, the *p*-value should be close to or above 0.9. Thermal diffusivity was then estimated by averaging the results of the 10 runs. For this assessment, the NLSF algorithm was used to determine the thermal diffusivity. The analysis revealed that, regardless of the standoff distance used, the estimated thermal diffusivity was the same up to two decimal points. This practice of conducting 10 repeats was then repeated for all the runs in the experiments.

Similar to the FEM analysis, it was important to compare the thermal diffusivity measurement using both Parker’s half-rise equation and the NLSF algorithm. Figure 10 shows the thermal diffusivity measurements using both methods at different energy levels. Similar to what is observed in Figure 4, Parker’s half-rise method seems to provide a slightly lower estimate for the thermal diffusivity at all thickness values and energy levels. Nonetheless, both methods follow a similar pattern for thicknesses above 4mm.

In terms of variation between the two methods based on the plots, it can be observed that at 4.8 kJ, the estimated values are closest to each other, indicating that this energy level may be optimum for thermal diffusivity measurement. The largest variation is observed within the plots obtained at 2.4 kJ of energy. The cause for this variation will be discussed in a later section. Based on the plots, several observations can be made. Firstly, the closeness of the estimated values using both methods across energy levels indicates the reliability of the estimated values. This result is consistent with expectations, as the NLSF algorithm continues to utilize the equation formulated by Parker. It is just not assuming the half-rise assumption that Parker developed on top of his equation for the temperature rise at the back wall. Secondly, the closeness in the measurement values also suggests that the methodology is robust, and therefore, the results were obtained following the same procedure that was outlined in the discussion section regarding the camera standoff distance. Thirdly, the estimated values are indicative of the true material thermal diffusivity. Compared to the value of 14 mm^2^/s provided by the supplier, the estimated values do cluster around this value. The only major anomaly in the results is the thermal diffusivity measurement for the sample at 5.933 mm, which was much higher (≈18 mm^2^/s). This high value for thermal diffusivity compared to the rest of the samples was observed in all the 10 repeats conducted for this sample. This anomaly is seen using both Parker and NLSF at all energy levels, so this does not reduce the reliability of the methodology; however, it is something that requires further investigation. Excluding this result, the maximum percentage difference from the supplier value is 13.27% obtained for the sample with thicknesses of 3.074 mm at 4.8 kJ flash energy. Finally, the plots show repeatability, as both methods estimate similar values for thermal diffusivity across different energy levels.

Figure 11a,b isolate the measurement methods and show the variation of thermal diffusivity measurements at the three different energy levels. The estimated values show a greater level of consistency at higher thickness values. For the NLSF method, the diffusivities estimated at 9.6 kJ and 4.8 kJ are in close agreement with a slightly higher variation at 2.4 kJ, where a higher value is estimated, especially for the 9.87 mm thick sample. This variation is not visible with Parker’s half-rise method, which is in line with the simulation results that indicate a greater level of accuracy in thermal diffusivity measurements at higher thicknesses. Overall, the NLSF method provides a higher level of repeatability, especially at higher energy levels, and both methods exhibit a similar trend for thermal diffusivity measurement.

The histograms in Figure 12 display the variation of thermal diffusivity using both methods at different energy levels. Both methods provide a bell distribution with the median value for all the runs to be between 15 mm^2^/s and 16 mm^2^/s. A closer review of the data, however, reveals that the range of thermal diffusivity values estimated using Parker at 4.8 kJ and 9.6 kJ is slightly larger (13–17 mm^2^/s) compared to NLSF (14–17 mm^2^/s). At 2.4 kJ, both methods offer the same level of variability in thermal diffusivity estimation. For NLSF, it is observed that the energy level has a negligible effect on the overall variability in thermal diffusivity estimation. A detailed analysis of the thermal diffusivity measurement is provided in Table 4. The term “CV” refers to the coefficient of variation, which is the mean divided by the standard deviation. This term measures the dispersion of data points in a data series around the mean.

The bottom row for each estimation method (NLSF and Parker’s) represents the overall mean of the thermal diffusivity values and their corresponding coefficient of variation (CV) from the rows above. Overall, the obtained CV using NLSF is much lower than the one obtained using Parker’s method, indicating that the precision of the NLSF is higher. For each individual thickness, the CV is lower at higher energy levels. A plausible explanation, as explained earlier, is that higher energy levels cause a greater temperature rise at the back wall. This means that the noise generated from the IR radiometer has less influence on the thermal diffusivity measurement. At lower energy levels, the back wall temperature will be lower. This means that the SNR is lower, which in turn causes a larger variation in the temporal plots of consecutive runs. In terms of the difference in the total mean at different energy levels, the means obtained using NLSF exhibit a greater degree of closeness compared to Parker’s method, once again building a stronger case for the use of NLSF for thermal diffusivity measurement as it offers greater repeatability. For the CV values, again, the NLSF displays a lower variation compared to Parker’s method. An additional consideration that must be taken here is that the CV values are higher due to the inclusion of the thermal diffusivity obtained from the sample that has a thickness of 5.933 mm. Removing this value will significantly reduce the CV for both methods at all energy levels. For this reason, the rest of the results have excluded these values. Figure 13 shows the Pareto charts for both the NLSF and Parker methods. For the NLSF, the influence of energy levels is insignificant, as seen in Figure 13a, and the thickness seems to dominate the thermal diffusivity measurement. This outcome is logically consistent, as the thermal diffusivity measurements exhibited greater variability with changes in thickness compared to the simulations. However, the change does not exhibit any identifiable pattern; i.e., it is random. The fluctuation of the thermal diffusivity with thicknesses can possibly be reduced by extending the full-factorial DOE by adding repeats of multiple samples with the same material and thickness. For Parker’s method, while both factors display statistical significance in measurement, the thickness seems to have a greater influence on the measurement than the energy level. A final comparison of the coefficient of variation is done in Figure 14, excluding the estimated values for the 5.933 mm-thick sample. Figure 14a shows the mean CV at different thicknesses averaged for all energy levels. Overall, NLSF offers a much lower CV, although the values are similar for both methods at higher energy levels. It is also observed that Parker’s method is more susceptible to inaccuracies at lower thickness levels, similar to what was observed in the FEM results. Once again, similar to Figure 11, the measurement methods are isolated to observe the effect of the energy levels at different thicknesses. It can be observed that at 9.6 kJ and 4.8 kJ, the SNR from the NLSF algorithm (Figure 14b) is sufficiently high to maintain a CV value of less than 2%, even at the highest thickness values, i.e., 9.87 mm.

The increased CV value at higher sample thicknesses and lower energy values can be attributed to the fact that this particular setting does not cause the back wall temperature to rise significantly. Combining the lower energy level with a higher sample thickness results in the back wall temperature signal being significantly corrupted by noise. Higher noise levels result in greater variability in thermal diffusivity estimates. For Parker’s method, the CV values are higher, especially at the upper and lower bounds of the thickness range. Unlike the NLSF method, a clear pattern cannot be observed, and the CV values are more randomly spread. Nevertheless, it can be concluded that Parker’s method yields more precise thermal diffusivity estimates for data captured at higher energy levels and for higher thickness values. A comparison between Figure 14a,b shows that the NLSF algorithm outperforms Parker, as it achieves a lower overall CV across all thicknesses and energy levels when estimating thermal diffusivity.

### 4.3. Comparison Between FEM and Laboratory Experiments

One of the research gaps highlighted by Ali et al. [19] was the limited number of FEA models that describe the back-wall temporal behaviour of a sample subjected to pulsed heating. One of the reasons for this could be the fact that the current models are not a true representative of real-life conditions since there are other factors that may cause disparities between simulation and laboratory experiments. Moreover, current FEA models have focused mainly on the reflection mode and the identification of defects, as shown by Kalyanavalli et al. [56]. The current literature does not contain many FEA models that describe temporal behaviour using the transmission mode. Furthermore, comparisons of the thermal diffusivity measurements using FEA and laboratory experiments are also extremely limited [56,57]. This research attempts to understand the disparity between FEA and laboratory experiments. For this purpose, Figure 15 was generated, which compares the thermal diffusivity measurement using both NLSF and Parker’s methods. For the comparison, the thermal diffusivity for all samples, excluding the 5.933 mm-thick sample, was considered. For the laboratory experiments, a total of 10 repeats were conducted for each thickness, with the displayed points representing the average of those runs. In contrast, repeats were not performed for the simulations, as the consistent nature of the simulation environment ensures identical results for the same input parameters.

Figure 15a,b shows that the results are far more consistent in the simulations compared to the laboratory experiments. Furthermore, there seems to be a systematic shift in the measurement for the laboratory experiments. This shift highlights several factors that have not been discussed in existing research within this context. These are

Planck’s law of blackbody radiation shows that there is an inverse relationship between the wavelength and the peak emission of electromagnetic radiation. This factor is not present in the simulation model as it applies pulse energy, assuming that it uniformly causes a temperature rise on the surface of the sample. This means that the energy distribution across wavelengths is assumed to be equal across different energy levels in the simulation model. In the laboratory experiments, as the energy of the flash reduces, the peak emission shifts to a longer wavelength. In our case, the IR radiometer has a fixed wavelength range (3–5 μm), so the captured radiation does not truly represent the spectral radiance of the body.As mentioned in the previous subsection, the SNR at lower energy levels is lower. This means that the noise levels are more prominent since the temperature increase at the back wall is much lower. While, theoretically, a low temperature increase should not affect the thermal diffusivity measurement as seen in the simulations, it does in laboratory experiments. This is because the temperature measurement systems, especially the IR radiometers’ (NETD), are a limiting factor. This factor determines the smallest temperature difference an IR radiometer can detect. At lower energy levels, the low temperature increase at the back wall can adversely affect the camera’s ability to capture accurate and meaningful thermal data. Nevertheless, based on Figure 15a, it can be observed that even at lower energy levels, i.e., 2.4 kJ, the NLSF algorithm still manages to provide thermal estimates that are closer to each other compared to Parker, further strengthening its thermal diffusivity estimation capabilities.Deviations can occur within the material composition itself. FEA models assume that the entire structure is homogeneous, which will result in repeatable results for intrinsic material properties such as thermal diffusivity. This is not the case for laboratory experiments, as demonstrated by our results, which showed variations in thermal diffusivity for the same material composition.Uncertainties may arise from repeated tests. In simulations, regardless of the number of repeats done, if input parameters are the same, the output (in this case, the temperature plots at the back wall) will be the same. However, in experimental conditions, it is nearly impossible to achieve such a result; hence, there is a disparity between runs, even for the same sample. The effect of the last two parameters has been documented previously in [34].Deviations can occur based on the heating pattern. For the simulations, a heat flux distributed uniformly across the sample is applied. However, for the laboratory experiments, two flash lamps are used to thermally excite the specimen. Unlike simulations, the heat distribution across the sample surface in laboratory experiments is uneven, which can negatively affect the accuracy of thermal diffusivity measurements. Moreover, while the energy level at the source of the flash lamps might be a certain value, the nominal value that is deposited at the surface would be significantly less, as demonstrated by Krankenhagen et al. [58].

It can be seen that the method used to determine material thermal diffusivity significantly influences the variability of the obtained values. As shown in Figure 15, thermal diffusivity estimates at 2.4 kJ exhibited greater variability when using Parker’s method compared to NLSF. In contrast, at 4.8 kJ and 9.6 kJ, the estimated values remained more consistent across all thicknesses. However, at the largest thickness (9.87 mm), Parker’s estimates are more consistent throughout all energy levels. The disparity in measurements using NLSF at lower energies and higher sample thicknesses could be attributed to its consideration of the entire back wall temperature curve rather than a single value, as in the Parker method, making it more susceptible to noise in the acquired signal, which translates to a higher disparity in the thermal diffusivity measurement. The reduced SNR due to lower excitation energy translates to a higher disparity in thermal diffusivity measurements. However, in terms of precision, the NLSF still outperforms Parker, especially at lower thicknesses and higher energy levels.

Figure 16 shows the residual plots of the thermal diffusivity measurements using both NLSF and Parker for the laboratory experiments. Comparing the plots to what was observed in the FEA model, similar conclusions can be drawn. Both FEA and laboratory experiments display normal distribution and show no indication of any bias that may have affected the measurement of thermal diffusivity. Furthermore, it can be seen that this behaviour is exhibited regardless of the method used to estimate thermal diffusivity. This demonstrates that the methodology employed in this research is a bias-free approach to thermal diffusivity measurement, effectively achieving the objective of establishing a simple step-by-step procedure that can be easily replicated for measurements in the transmission mode. Contrary to what is observed in the FEA analysis (Figure 9), comparing the residual values in Figure 16a,b shows that the variability in the estimated values is halved when the NLSF algorithm is chosen to evaluate thermal diffusivity (0.5 using NLSF and 1.0 using Parker’s half-rise method). This implies that the NLSF algorithm offers more precise estimates for thermal diffusivity. The FEA model plots display lower residual values compared to the laboratory experiments, which is expected due to the reasons previously mentioned. Moreover, in the simulations, both methods display similar values for the residual.

## 5. Conclusions

While the through-transmission measurement technique for PT has been around for a long time, it has not seen significant developments compared to the reflection mode. A major obstacle hindering the technique’s development is its inability to quantify defect depth, even though it is able to detect deeper defects compared to the reflection mode. Moreover, there is a notable absence of theoretical models that accurately describe back-wall temperature behaviour. The literature also lacks a step-by-step guide for conducting through-transmission thermography measurements, especially in the laboratory setting. This study addresses these challenges by employing a full-factorial DOE approach, combining FEA and laboratory experiments to evaluate thermal diffusivity. Thermal diffusivity is estimated using two methods: the NLSF and Parker’s half-rise equation. The ability to accurately estimate thermal diffusivity and corresponding measurement uncertainties is compared. Overall, a simple and repeatable methodology that can be easily replicated in laboratory settings without the use of expensive commercial equipment is presented. Moreover, the introduction of multiparameter optimisation using the NLSF method allows for a more accurate estimation of thermal diffusivity as it removes some of the assumptions for certain parameters that Parker makes. One of these assumptions is that the reflection coefficient *R* = 1, which might not always be the case in a practical setting. While these parameters are difficult to measure in real life, they are crucial in accurately assessing the thermal behaviour of materials. The implementation of DOE is crucial because, although the proposed NLSF algorithm provides more accurate and repeatable results, it is essential to demonstrate that these results are unbiased. The key findings of this research are listed below:Parker’s method exhibits higher levels of accuracy when estimating material thermal diffusivity at higher thicknesses. In contrast, the NLSF algorithm offers similar levels of measurement accuracy regardless of the material thickness (see Section 4.1).The NLSF algorithm is a more robust method for estimating material thermal diffusivity compared to Parker’s method. This is demonstrated by adding noise to the temperature signal obtained from the FEA model. At extremely low SNRs, i.e., −20dB, the NLSF offered a significantly lower measurement error value (≈60%) compared to Paker (≈1600%) (see Section 4.1).The NLSF algorithm demonstrated repeatable results (CV < 4.3%) compared to Parker’s method (CV ≈ 7%). Additionally, the NLSF exhibited a positive correlation with increasing excitation energy in terms of measurement repeatability, whereas Parker displayed no such trend (see Section 4.2).From this research, several key factors such as NETD, material homogeneity, and measurement repeatability influence the thermal diffusivity measurement in the real world as opposed to simulations. These factors are not considered in the FEA model, which has led to disparities in thermal diffusivity measurement, as demonstrated by the comparative analysis of FEA and laboratory experiments (see Section 4.3).Finally, it has been observed in experimental conditions that the choice of algorithm affects the variability in the thermal diffusivity estimates. The variability in thermal diffusivity estimates is halved by choosing NLSF instead of Parker. However, this effect is not present in the simulations (see Figure 9 and Figure 16). This makes the NLSF algorithm a more attractive candidate for conducting thermal diffusivity measurements.

These findings are crucial to obtaining a comprehensive understanding of through-transmission thermography for thermal diffusivity measurements. Furthermore, the insights from the comparative analysis between FEA and laboratory experiments pave the way for a potential correction factor that can be implemented to reduce the current disparity. However, due to the volume of results presented in the study, the implementation of a correction factor has been planned for a future study. The incorporation of a correction factor in simulation models can significantly enhance the accuracy of material thermal behaviour modelling, enabling more effective prognostics and preventive strategies to mitigate failure. This advancement could lead to a substantial reduction in resource consumption and waste generation, thereby contributing to a more sustainable and environmentally friendly approach within the industry.

## Figures and Tables

**Figure 1 sensors-25-01139-f001:**
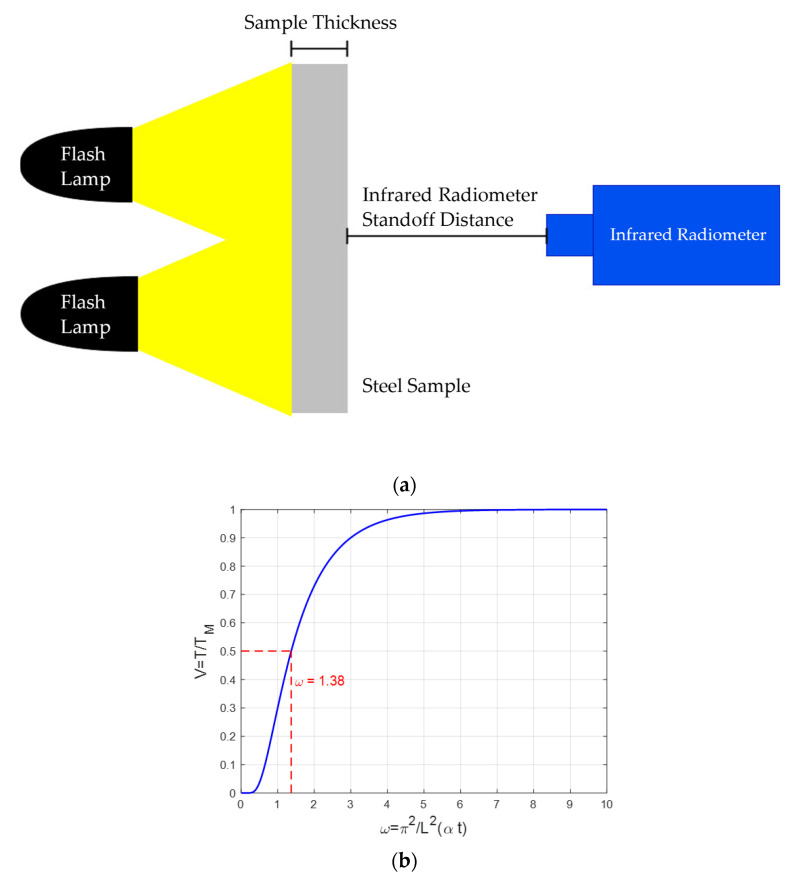
(**a**) Illustration of the PT transmission mode configuration; (**b**) nondimensionalised back wall temperature plot displaying the value for ω at half the maximum temperature.

**Figure 2 sensors-25-01139-f002:**
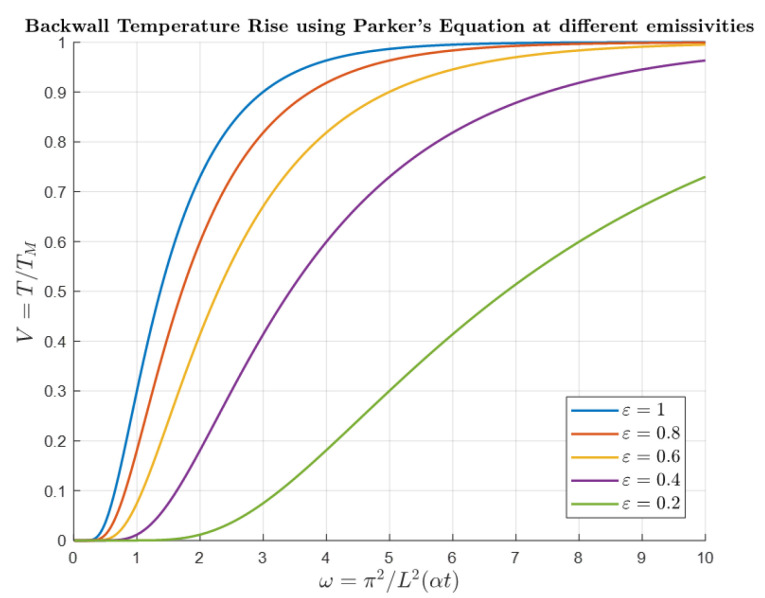
Normalised back wall temperature profile at different emissivity values.

**Figure 3 sensors-25-01139-f003:**
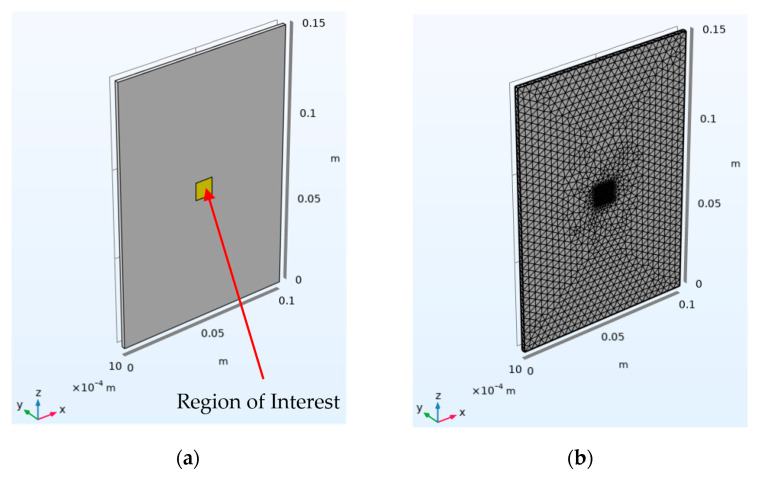
Geometry for the FEA Model. (**a**) The yellow region is where the temperature measurements are taken from at the back wall. (**b**) The meshed geometry is where the region of interest has a finer geometry to save computational time.

**Figure 4 sensors-25-01139-f004:**
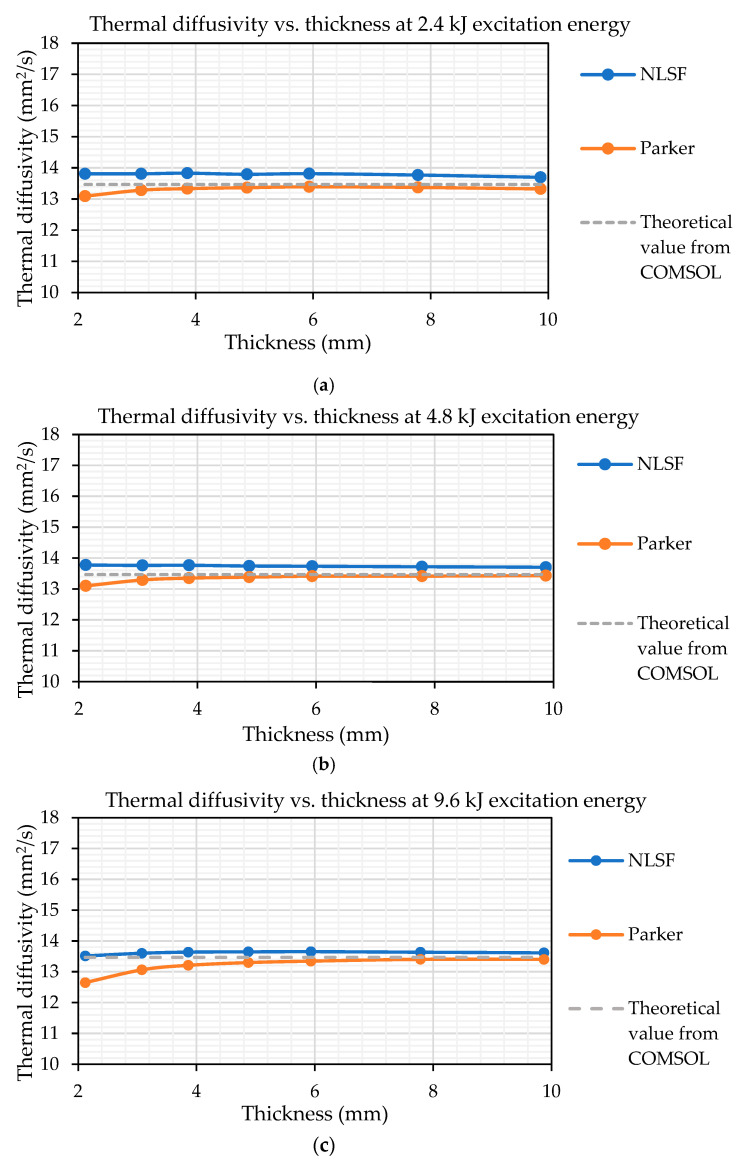
Thermal diffusivity measurements using the NLSF algorithm and Parker’s method for a 150 × 100 mm S275 steel plate with multiple thicknesses, subjected to different energy levels of pulsed heating simulated in COMSOL. Thermal diffusivity estimates at (**a**) 2.4kJ (**b**) 4.8kJ and (**c**) 9.6kJ.

**Figure 5 sensors-25-01139-f005:**
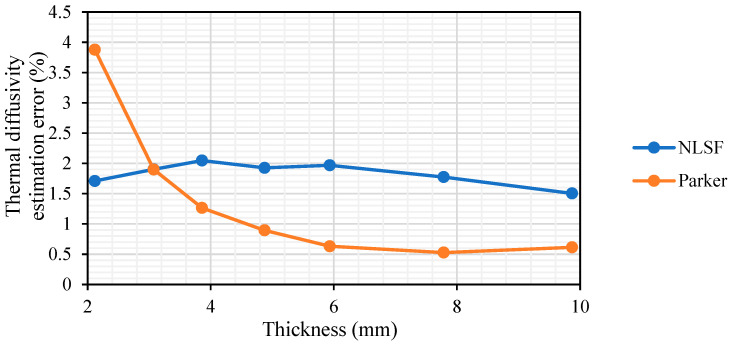
Measurement error for NLSF and Parker’s method across all thicknesses (values have been averaged across the three different energy levels).

**Figure 6 sensors-25-01139-f006:**
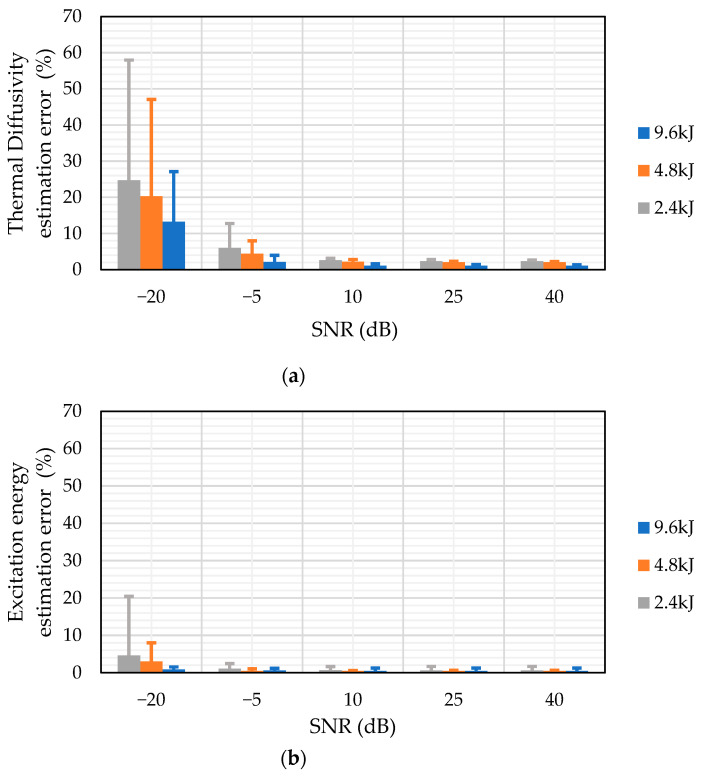
Error analysis of the NLSF algorithm with varying SNRS for (**a**) thermal diffusivity measurement and (**b**) input energy measurement using the temperature plots obtained from the FEA model in COMSOL.

**Figure 7 sensors-25-01139-f007:**
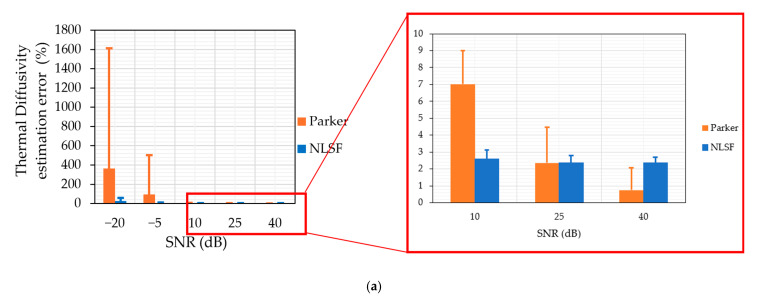
Comparison of Parker’s and NLSF method’s performance at estimating thermal diffusivity at different SNRs for energy levels (**a**) 2.4 kJ, (**b**) 4.8 kJ, and (**c**) 9.6 kJ.

**Figure 8 sensors-25-01139-f008:**
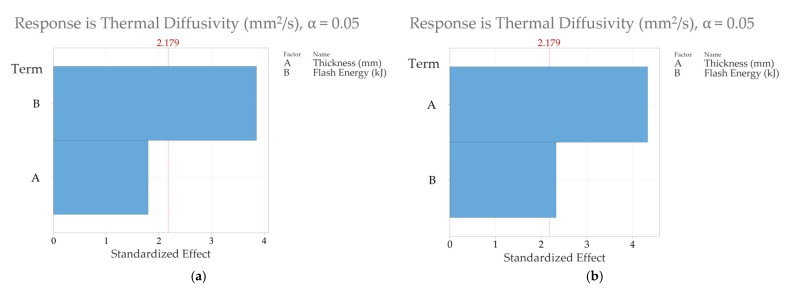
Pareto charts for standardised effects for the finite element simulations (**a**) thermal diffusivity estimated using NLSF and (**b**) thermal diffusivity estimated using Parker’s method.

**Figure 9 sensors-25-01139-f009:**
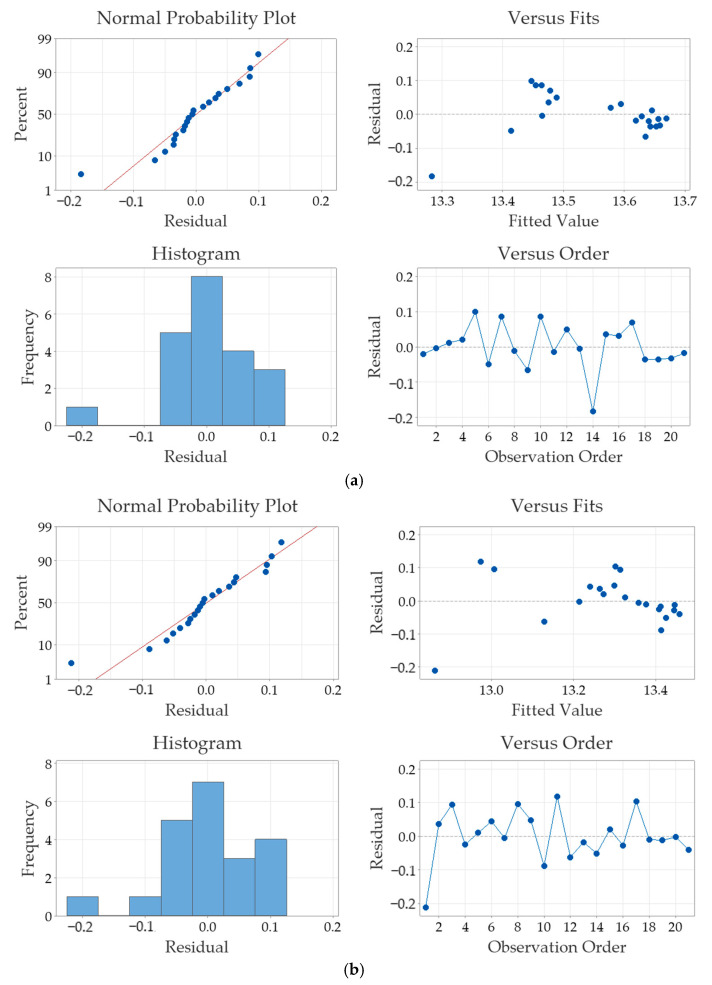
Residual plots from the finite element simulations for estimating thermal diffusivity using (**a**) NLSF and (**b**) Parker’s method.

**Figure 10 sensors-25-01139-f010:**
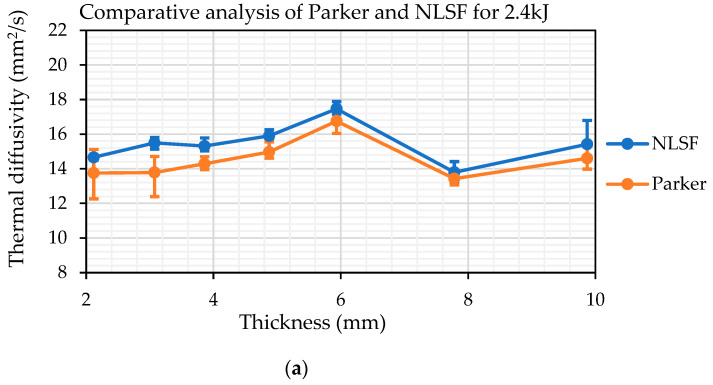
Thermal diffusivity estimations for different thicknesses at (**a**) 2.4kJ, (**b**) 4.8kJ, and (**c**) 9.6kJ using NLSF and Parker’s methods.

**Figure 11 sensors-25-01139-f011:**
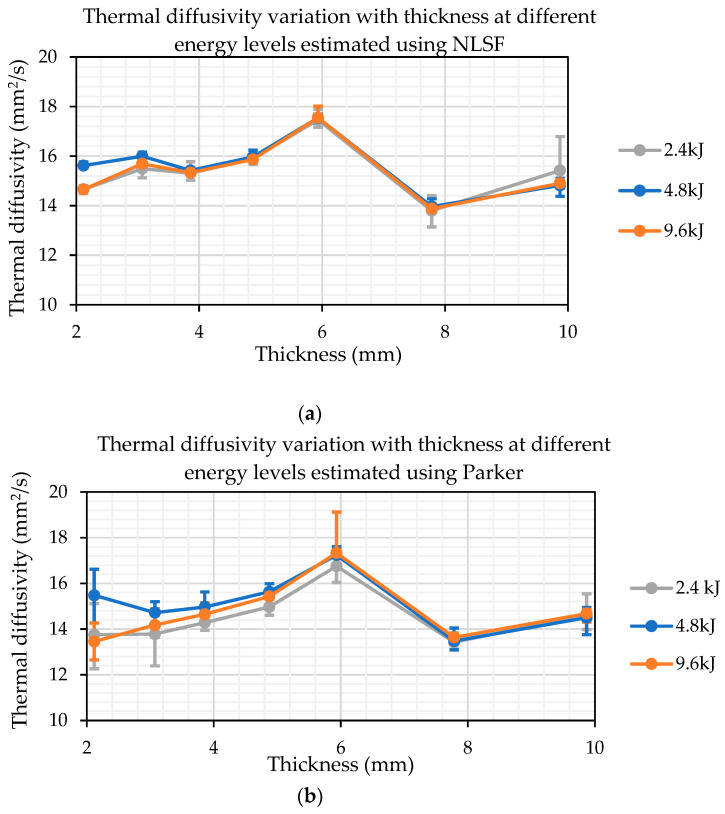
Thermal diffusivity measurements for varying thicknesses at different energy levels, estimated using (**a**) NLSF and (**b**) Parker’s method.

**Figure 12 sensors-25-01139-f012:**
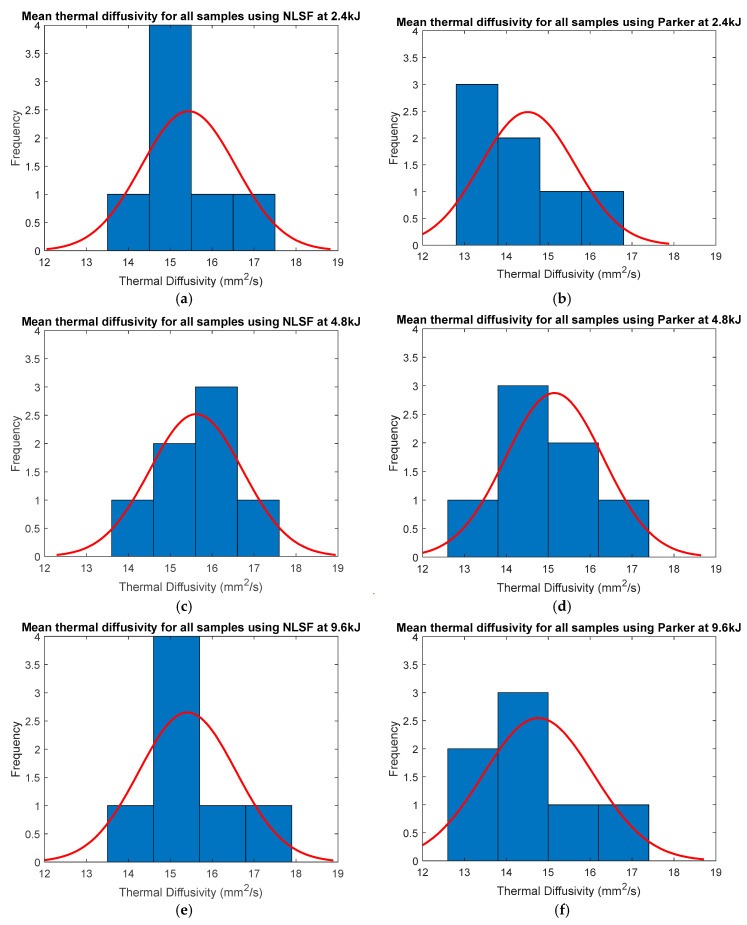
Histograms showcasing variations in thermal diffusivity with different thicknesses at different energy levels estimated using NLSF (**a**,**c**,**e**) and Parker’s methods (**b**,**d**,**f**).

**Figure 13 sensors-25-01139-f013:**
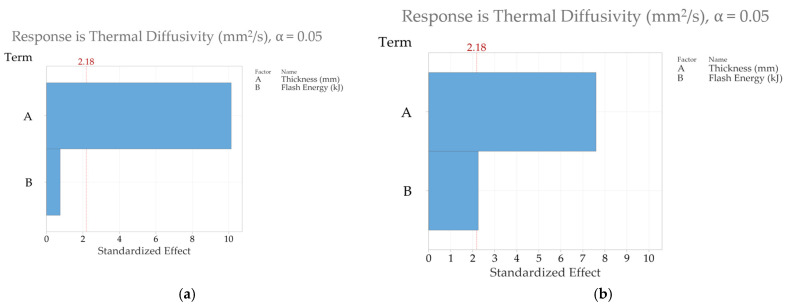
Pareto charts for standardised effects for the laboratory experiments (**a**) thermal diffusivity estimated using NLSF and (**b**) thermal diffusivity estimated using Parker’s method.

**Figure 14 sensors-25-01139-f014:**
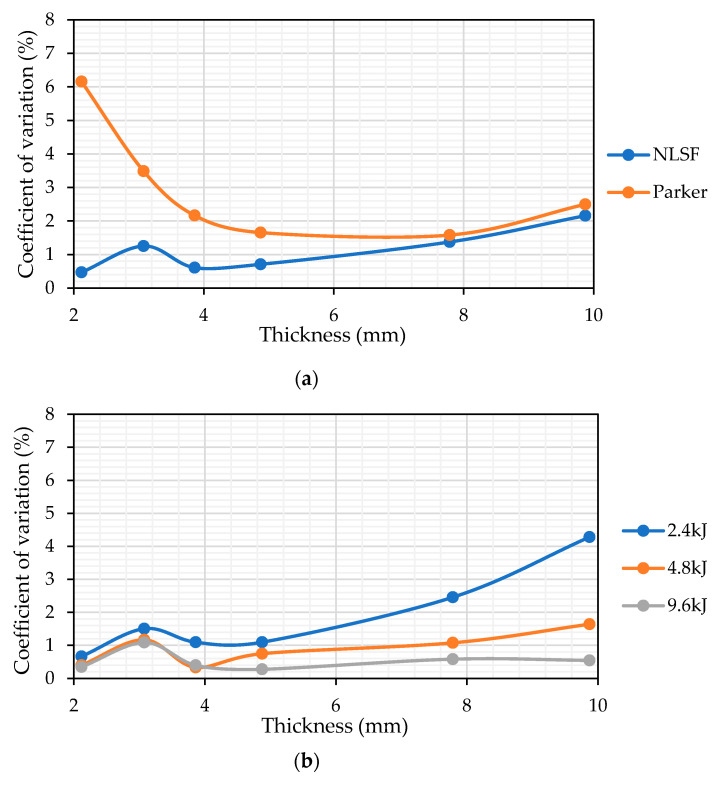
Comparison of the coefficient of variation (**a**) comparison between the NLSF and Parker methods, (**b**) variation at different thicknesses using the NLSF method, and (**c**) variation at different thicknesses using Parker’s method.

**Figure 15 sensors-25-01139-f015:**
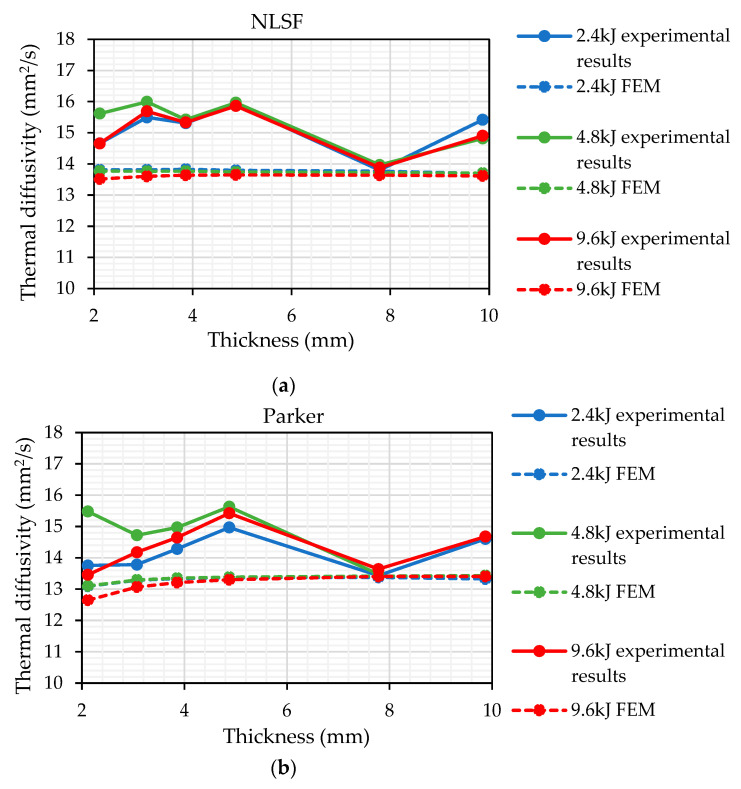
Comparison between simulation and laboratory experiments for thermal diffusivity measurement (**a**) using the NLSF method and (**b**) using Parker’s method.

**Figure 16 sensors-25-01139-f016:**
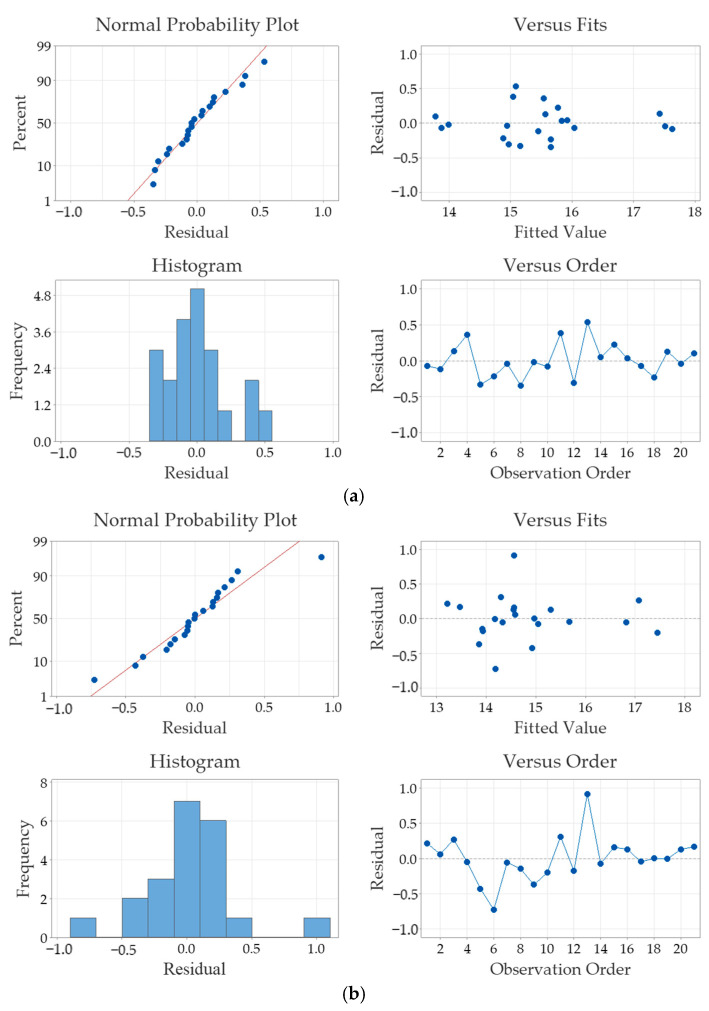
Residual plots from laboratory experiments for estimating thermal diffusivity using (**a**) NLSF and (**b**) Parker.

**Table 1 sensors-25-01139-t001:** Input parameters for the FEA analysis.

Material Thermal Properties	
**Specific Heat Capacity (J/kg K)**	470
**Density (kg/m^3^)**	7900
**Thermal Conductivity (W/m K)**	50
Plate Geometry	
**Height (mm)**	150
**Width (mm)**	100
**Thickness (mm)**	2.118, 3.074, 3.859, 4.874, 5.933, 7.784, 9.87
Heat Transfer Physics	
**Initial Temperature (K)**	293.15 (applied to the entire body)
**Pulse Energy (kJ)**	9.6, 4.8, 2.4
**Pulse Width (s)**	1/175, 1/300
**Heat Flux q_o_ (W/m^2^)**	((Pulse Energy/Pulse Width)/(Width × Height)) × B (t)
**Convective Heat Transfer Coefficient *h* (W/m^2^ K)**	5
**Emissivity ε**	0.95
Meshing Parameter (area of interest)	
**Mesh Size**	Extremely Fine
**Maximum Element Size (m)**	0.001
**Minimum Element Size (m)**	0.00001
**Maximum Element Growth Rate**	1.3
**Curvature Factor**	0.2
**Resolution of Narrow Regions**	1
Meshing Parameter (rest of the geometry)	
**Mesh Size**	Normal
**Maximum Element Size**	0.015
**Minimum Element Size**	0.0027
**Maximum Element Growth Rate**	1.5
**Curvature Factor**	0.6
**Resolution of Narrow Regions**	0.5

**Table 2 sensors-25-01139-t002:** Full-factorial DOE for flash thermography.

Runs	Flash Energy (kJ)	Sample Thickness (mm) (Measurement Uncertainty ± 0.008 mm)
1	2.4	3.859
2	4.8	7.784
3	9.6	5.933
4	9.6	4.874
5	2.4	2.118
6	4.8	4.874
7	9.6	7.784
8	2.4	4.874
9	2.4	5.933
10	4.8	3.859
11	9.6	9.870
12	4.8	3.074
13	2.4	7.784
14	9.6	3.074
15	9.6	3.859
16	4.8	2.118
17	2.4	9.870
18	4.8	9.870
19	4.8	5.933
20	9.6	2.118
21	2.4	3.074

**Table 3 sensors-25-01139-t003:** Error analysis of the NLSF algorithm estimated using the temperature plots obtained from the FEA model in COMSOL with no noise.

Input flash energy 2.4 kJ							
**Thickness (mm)**	**Thermal diffusivity (mm^2^/s)**	**Error (%)**	**Estimated energy input (kJ)**	**Error (%)**	**Reflection coefficient**	**Error (%)**	**Sampling time *t_s_* (μs)**
2.118	13.8109	2.530809	2.3961	0.162500	−1.0300	3.00	1.1208
3.074	13.8115	2.535264	2.3973	0.112500	−1.0217	2.17	1.7943
3.859	13.8301	2.673348	2.3985	0.062500	−1.0201	2.01	2.6883
4.874	13.7948	2.411284	2.3872	0.533333	−1.0175	1.75	3.8413
5.933	13.8148	2.559762	2.4112	0.466667	−1.0180	1.80	5.8116
7.784	13.7691	2.220490	2.4328	1.366667	−1.0165	1.65	9.4030
9.870	13.6971	1.685969	2.4409	1.704167	−1.0150	1.50	13.9650
Average	13.7897	2.373847	2.4091	0.629762	−1.0198	1.98	5.5178
Standard deviation	0.04511	0.334926	0.02032	0.650763	0.005003	0.50	4.6646
Input flash energy 4.8 kJ							
**Thickness (mm)**	**Thermal diffusivity (mm^2^/s)**	**Error (%)**	**Estimated energy input (kJ)**	**Error (%)**	**Reflection coefficient**	**Error (%)**	**Sampling time *t_s_* (μs)**
2.118	13.7747	2.262064	4.7765	0.489583	−1.0252	2.52	1.003
3.074	13.7641	2.183370	4.7952	0.100000	−1.0190	1.90	1.6299
3.859	13.7682	2.213808	4.7778	0.462500	−1.0168	1.68	2.3616
4.874	13.7435	2.030438	4.8272	0.566667	−1.0146	1.46	3.3930
5.933	13.7348	1.965850	4.8287	0.597917	−1.0140	1.40	4.8966
7.784	13.7203	1.858203	4.8309	0.643750	−1.0136	1.36	8.2483
9.870	13.7041	1.737936	4.8288	0.600000	−1.0126	1.26	12.4550
Average	13.74424	2.035953	4.8093	0.494345	−1.01654	1.65	4.8549
Standard deviation	0.02636	0.195723	0.02520	0.185287	0.004387	0.44	4.1411
Input flash energy 9.6 kJ							
**Thickness (mm)**	**Thermal diffusivity (mm^2^/s)**	**Error (%)**	**Estimated energy input (kJ)**	**Error (%)**	**Reflection coefficient**	**Error (%)**	**Sampling time *t_s_* (μs)**
2.118	13.5152	0.335561	9.4808	1.241667	−1.0491	4.91	1.8262
3.074	13.6030	0.987379	9.5483	0.538542	−1.0269	2.69	2.1679
3.859	13.6393	1.256867	9.5338	0.689583	−1.0204	2.04	2.7556
4.874	13.6499	1.335561	9.5669	0.344792	−1.0161	1.61	3.6349
5.933	13.6558	1.379362	9.5569	0.448958	−1.0140	1.40	4.8639
7.784	13.6376	1.244246	9.5972	0.029167	−1.0113	1.13	7.2157
9.870	13.6163	1.086117	9.6157	0.163542	−1.0100	1.00	10.6360
Average	13.6167	1.0893	9.5571	0.493750	−1.0211	2.11	4.728
Standard deviation	0.04842	0.3594	0.04393	0.397661	0.01361	1.36	3.1930

**Table 4 sensors-25-01139-t004:** Mean thermal diffusivities for different thicknesses at different energy levels and their corresponding coefficient of variations estimated using NLSF and Parker’s methods.

NLSF						
Thickness (mm)	Mean Diffusivity at 2.4 kJ (mm^2^/s)	CV (%)	Mean Diffusivity at 4.8 kJ (mm^2^/s)	CV (%)	Mean Diffusivity at 9.6 kJ (mm^2^/s)	CV (%)
2.118	14.66085	0.66614	15.61565	0.40183	14.65690	0.35052
3.074	15.49381	1.48823	15.99453	1.16981	15.68935	1.08882
3.859	15.31025	1.13897	15.4192	0.33212	15.33197	0.39875
4.874	15.89901	1.09679	15.9658	0.75168	15.86272	0.27864
5.933	17.46808	1.21517	17.54384	0.60735	17.55430	0.96350
7.784	13.79977	2.46065	13.96800	1.07911	13.87974	0.58508
9.870	15.41607	4.28442	14.81869	1.64175	14.90288	0.54605
Overall mean	15.43541	7.311262	15.61796	7.09755	15.41112	7.51748
**Parker**						
**Thickness (mm)**	**Mean Diffusivity at 2.4 kJ (mm^2^/s)**	**CV (%)**	**Mean Diffusivity at 4.8 kJ (mm^2^/s)**	**CV (%)**	**Mean Diffusivity at 9.6 kJ (mm^2^/s)**	**CV (%)**
2.118	13.75339	7.03374	15.47843	6.75203	13.45826	4.69093
3.074	13.78199	4.45796	14.72094	2.69299	14.17641	3.30880
3.859	14.28432	2.07046	14.96843	2.43636	14.64702	1.99543
4.874	14.96963	1.99153	15.63109	1.31188	15.42571	1.66790
5.933	16.75792	3.16088	17.24660	1.61679	17.33432	3.74955
7.784	13.43247	1.85876	13.47974	2.13000	13.63782	0.75683
9.870	14.60420	3.95801	14.50244	2.49668	14.67653	1.03334
Overall mean	14.51199	7.74913	15.14681	7.69978	14.76515	8.90992

## Data Availability

The raw data used for this study can be found at https://doi.org/10.57996/cran.ceres-2640.

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
