# Peer review of "Effective Thermal Diffusivity Measurement Using Through-Transmission Pulsed Thermography: Extending the Current Practice by Incorporating Multi-Parameter Optimisation"

_sensors, 2025, doi:10.3390/s25041139_

Round 1
Reviewer 1 Report
Comments and Suggestions for Authors
1. In Section 2.1 discussing Pulsed Thermography (PT), the authors mention its advantages such as fast, non-contact, and non-intrusive inspection. While these benefits are well-known, the manuscript lacks a comprehensive comparison with other thermographic techniques, especially in the context of material characterization. It would be insightful to see a detailed analysis of how PT outperforms or complements methods like Lock-in Thermography or Step Thermography, especially in terms of accuracy and repeatability.
2. In Section 6, the authors discuss quantization noise and its impact on thermal diffusivity measurement error. While this analysis is important, the manuscript does not explore other potential sources of error, such as environmental noise, thermal losses, or reflections from sample surfaces. A more comprehensive error analysis considering all significant error sources would strengthen the manuscript's conclusions.
3. The conclusions in Section 5 mention the lack of a transparent methodology for conducting through-transmission thermography measurements in the literature. While the authors propose a full factorial DOE (Design of Experiments) approach combining FE analysis and laboratory experiments as a solution, the manuscript does not fully demonstrate the effectiveness of this approach. Additional experimental results or case studies showcasing the application of the proposed methodology would significantly enhance the manuscript's impact and validity.
4. I believe it is crucial to compare the proposed method more comprehensively with the reflection mode, which is currently the more prevalent technique. I suggest conducting a detailed analysis that directly compares the through-transmission mode with the reflection mode under identical or similar experimental conditions. This comparison should include metrics such as defect detection limits, measurement error rates, and repeatability.
Reviewer 2 Report
Comments and Suggestions for Authors
In the propose papers, authors propose a cost-effective method for measuring thermal diffusivity using pulsed thermography, combining Finite Element Analysis (FEA) and physical experiments. Sample thickness and flash energy are mainly investigated. Thermal diffusivity is evaluated with two approaches: Parker’s half-rise equation and the New Least Squares Fitting (NLSF) algorithm and comparison is done.
Before considering the paper for a publication in Sensors, it must be improved and I added the following comments.
1)The abstract should be more concise to express the main novelty/outcomes of the propose methods. The terms ‘transparent’ methodology is not clear: what authors wants to highlight for the proposed methodology? In the abstract, it is mentioned that NLSF algorithm is preferred: I would suggest to quantify the parameters so that the reader can have a fast idea of the comparison between the two method in the abstract.
Moreover, some terms are abbreviated, others are not (pulsed thermography, for example): authors should check them.
2) In the introduction, several methods are mentioned to show that the proposed approach, based on transmission mode, has advantages. As in the abstract, the introduction sounds dispersive. For example, in line 98-100 authors explain their work: “This work attempts to apply 98 some of the potential solutions described in that study to help overcome some of the cur-99 rent challenges impeding the progress of through-transmission thermography.”: I suggest to review this paragraph. Did authors referred to their own work? Or a citation? If referred to their own work, then is recommended to compact the explanation of their research outcomes in the last paragraph of the introduction. Moreover, terms and abbreviations should be reviewed, for example some abbreviations are not used in the text, some of them are repeated (as example: additive manufacturing (AM) is repeated twice). The use of many abbreviations, where not necessary, should be avoided to help the reader in understanding the introduction.
3)The study is focused on two parameters only: thickness and flesh energy. It is not clear the motivation behind this choice. Are there other parameters that are negligible? How the values of thickness and flesh energy are chosen in the model?
4)Authors declared that the novelty of this paper lies in assessing the viability of thermal diffusivity measurement of using PT in the through-transmission configuration using multi-objective optimisation. As multi-object, are authors referred to the investigated parameters (thickness, flesh energy)? If yes, why two parameters only were chosen and what is the optimum number of parameters for the model to be investigated.
5) Figure 1 a) and b) must be improved. The setup is not clear and it is not related to the section in which it is inserted. What is the sample investigated? The camera, the source? All these features are not clear. The Graphs are not labelled with the measurement units, making them difficult to understand.
6) Same comment as Figure 1 for Figure 2: axes are not clear, the font is too small.
7) The region of interest is selected at the center of the plate. IS there a reason why authors selected the center? It would be interesting to evaluate the effect in different regions.
8) In table 1, model’s parameters are reported. It is not clear the choice of some important parameters such as depth. The explanation should be included in the manuscript. Moreover, in the table is reported a mesh: ‘extremely fine’: this parameter should be quantified, since the mesh strongly affect results.
9)The graphs of the results (from Figure 3) should be improved. Font should be improved, how the graphs are displayed, grid ox the axes.
10)Table 3 reports the results of the analysis. Standard deviations, average values or other statistical parameters should be considered for a comparison of the results.
11)The zoom figures in Figure 6 must be explained and reported in the description of the Figure. Moreover, the figures must be improved.
12)Experimental results are shown in Figure 14. How many data points? From the Figure, seems that only 2 data points are reported. Is that correct? How many repetition of the experiment? Two data points are not enough for an evaluation and comparison with the mode. Moreover, lab measurement is not a term that should be used in a scientific publication. I suggest to refer to measurements as experimental measurements/results.
Comments on the Quality of English Language
English should be improved to obtain a more concise text, in particular for abstract and introduction
Round 2
Reviewer 1 Report
Comments and Suggestions for Authors
I have no further question
Author Response
Comment 1: I have no further question
Response 1: Dear Reviewer, thank you for taking the time to review our manuscript and providing us with valuable feedback. The authors are glad that we have been able to answer all your queries to your satisfaction.
Reviewer 2 Report
Comments and Suggestions for Authors
I carefully read the responses and the reviewed version of the propose manuscript.
1) Figures should be still improved. They appear in the text in the wrong position and for a nbetter visualization, the legend should be included in the figure.
2) Figure 1 is poor of content: text is not clearly visible, it is not explaned the view (top/longitudinal/etc). This figure in particular with the setup should be improved.
3) Authors modify Figure 15 adding more points. What is the reason of the non linear behaviour in the terminal diffusivity as a function of thickness?
4) Since the manuscript contains many information in terms of statistical parameters, it would be helpful to have, where possible, figures that summarize the results. Table with calculation could be included in a supplementary section.
5) References must be checked. Some references have DOI, others are reported without doi. I strongly recommend the authors to carefully review the references according to the suggested template.
Comments on the Quality of English Language
English improved.
